# Conserving Refugia: What Are We Protecting and Why?

Maurizio Rossetto [1,2,*] and Robert Kooyman [1,3]

1  Research Centre for Ecosystem Resilience, Australian Institute of Botanical Science, The Royal Botanic, Garden Sydney, Sydney, NSW 2000, Australia; robert.kooyman@mq.edu.au
2  Queensland Alliance of Agriculture and Food Innovation, University of Queensland, Santa Lucia, QLD 4072, Australia
3  Department of Biological Sciences, Macquarie University, Sydney, NSW 2109, Australia
*  Correspondence: maurizio.rossetto@rbgsyd.nsw.gov.au

**Abstract:** Refugia play an important role in contributing to the conservation of species and communities by buffering environmental conditions over time. As large natural landscapes worldwide are declining and are increasingly threatened by extreme events, critical decision-making in biological conservation depends on improved understanding of what is being protected by refugia and why. We provide three novel definitions for refugia (i.e., persistent, future, and temporary) that incorporate ecological and evolutionary dynamics into a land management decision framework and are applicable across changing temporal and spatial settings. Definitions are supported by identification, core value, and management strategy criteria to assist short- and long-term decision-making. We illustrate these concepts using the World Heritage Gondwana Rainforests (WHGR) of eastern Australia, briefly exploring the spatial and temporal factors that can inform the development of conservation management strategies following the extreme fire events of 2019–2020. For the WHGR, available knowledge can be used to protect critical assets by recognizing and implementing buffer zones and corridor connections, and by undertaking emergency translocations of target species into safe areas that will act as future refugia. More broadly, we suggest that the identification and protection of ecological and evolutionary processes across varying temporal and spatial scales is central to securing improved biodiversity conservation outcomes.

**Keywords:** biodiversity conservation; landscape management; environmental buffering; persistent refugia; future refugia; temporary refugia

## 1. Introduction: Refugia as Conservation Priorities

Protected areas provide in situ persistence and survival opportunities in extreme conditions and in times of change [1,2]. Refugia, by definition, play an important and ongoing role in contributing to the survival and conservation of species and communities by buffering environmental conditions through time [3] and across scales, ranging from microrefugia [4] to continental biodiversity hotspots [5]. Here, we focus on the functional aspects of areas protected from, or buffered against, regular disturbance and change, while contributing to the natural dynamics of the regional community assembly processes as well as biodiversity conservation targets and outcomes [3,6,7].

Globally, large intact natural landscapes are declining, and areas of significant diversity and evolutionary value that include refugia are increasingly threatened by extreme events [8]. Subsequently, the establishment of effective biodiversity protection mechanisms is essential to secure long-term conservation outcomes [9]. For refugia to be incorporated as part of global biodiversity protection strategies, it is critical to understand the evolutionary history and ecological processes that they protect and the type, likelihood, and nature of threats to their dynamic stability. To support conservation management, we need to identify what is being protected by refugia, and to characterize them across temporal and spatial scales.

The scale of the Australian fires of 2019–2020 provide an example of the growing challenges associated with protecting and conserving refugia in the face of extreme events, and how appropriate responses can be hampered by a lack of understanding of the role of refugia and why they are important [10]. Our objective here is to provide a set of definitions for refugia that incorporate ecological and evolutionary dynamics into a management decision framework placed within ever-changing temporal and landscape-level scenarios. Previous frameworks for identifying refugia have advocated for the inclusion of functional traits and the phylogenetic structure of community assemblages (e.g., [11]). We provide a broader range of definitions for refugia that nest those components into a dynamic framework, representing different temporal and spatial configurations of refugia. We then illustrate the concepts with a specific example, the World Heritage Gondwana Rainforests of eastern Australia (WHGR), briefly working through relevant definition and management criteria to explore the spatial and temporal issues needed to inform the development of conservation management strategies in disparate contexts.

## 2. Management-Oriented Definitions of Refugia

From an evolutionary perspective, large-scale spatial patterns of biodiversity depend on three variables: diversification (origination) rates, extinction rates, and changes in geographic distributions [12]. From an ecological perspective, the traits that shape persistence, dispersal, abundance, competition, and regeneration influence differently each species' survival and community assembly role in relation to environmental gradients [13]. In order to better represent varying temporal scales and landscape settings, we characterize refugia into three types: persistent, future and temporary (Table 1). While we acknowledge the inevitable chance of overlaps in location (both spatially and temporally), we suggest that framing refugia within the proposed context of dynamic stability is critical to the improvement of conservation management in continuously changing circumstances [3] and in relation to increasing anthropogenic impacts [14].

The criteria used to first identify refugia and then characterize the key (core) values of such areas are listed in Table 1. The identification criteria and core values provide the basis for the development of both management strategies and important conservation policy settings.

### 2.1. Persistent Refugia

Persistent refugia can be defined as geographical locations of continuous occupation that represent the maximum range contraction of a species or a vegetation community due to habitat filtering [15,16]. At longer biogeographic timescales, persistent refugia can also be established through habitat tracking across vast geographic distances [17], suggesting that they not only retain a diversity of taxa and lineages but they also protect and promote biotic interactions and adaptations to stable environments representative of ancient epochs [18–20]. Persistent refugia offer continuity (persistence) in specific environmental conditions on evolutionary time scales [21], and often protect landscape heterogeneity and preserve diverse faunas and floras that can include ancient lineages and threatened species [22]. In turn, the long-term preservation of evolutionary history and evolutionary potential also establishes opportunities for speciation and specialization through lineage expansion onto new areas.

**Table 1.** Summarized definitions of the three types of refugia described, i.e., why they are important, how to identify them, and what the relevant key management strategies are. The identification and core criteria provide the basis for the development of management strategies and conservation policy settings.

| | Persistent Refugia | Future Refugia | Temporary Refugia |
|---|---|---|---|
| Main Identification Criteria | • Distinctiveness at compositional, functional, phylogenetic, and genetic levels.<br>• Community assembly processes and persistence based on paleo-ecological and fossil evidence.<br>• Signals of contraction/expansion dynamics using refined landscape-genomic and phylogeographic data capturing the spatial distribution of diversity.<br>• Areas of high environmental stability inferred from species distributions in environmental niche models (ENMs). | • Areas of future environmental stability inferred from species distributions in environmental niche models (ENMs), which are likely to be suitable for on site management or to source material for translocation.<br>• Informed by innovative research on species' adaptive capacity that combines physiological, ecological, and genetic information, within the context of land use projections. | • Refugial habitats of varying size that survive in larger areas subjected to stochastic extreme events.<br>• Specific localized conditions that buffer areas from extreme events (e.g., fire-proof pockets protected by streams, cliffs, or scree slopes).<br>• In a larger landscape context, localized conditions can also protect spatially larger areas within 'wilderness areas' that in some scenarios may represent whole National Parks or Nature Reserves. |
| Core Value Criteria | • Stable environmental conditions across evolutionary time scales, resulting in continuous occupation of a species or a vegetation community in a location.<br>• Alternatively, can include habitat tracking to emerging refugia across biogeographic timescales.<br>• Emphasis is on species that co-occur as a consequence of environmental conditions, ecological attributes (e.g., traits), and evolutionary processes. | • Areas predicted to be buffered from anthropogenic climate change allowing for relative (dynamic) stability into the future, and with secure land tenure. | • Areas that have not been impacted by extreme disturbance events surrounding them.<br>• Areas with a critical role in favoring localized recovery and recruitment of local fauna and flora post disturbance of surrounding impacted areas.<br>• Ex situ 'refuges' established through translocations or represented by living collections can represent a last resort for species or for targeted genetic resources. |

**Table 1.** *Cont.*

| | Persistent Refugia | Future Refugia | Temporary Refugia |
|---|---|---|---|
| Management Strategy Criteria | <ul><li>Areas of high biodiversity value, safeguarding unique ecological history, and evolutionary potential that need to be protected through legislative action.</li><li>Management plans need to include active protection and buffering from disturbances that could upset existing balances.</li><li>Does not exclude indigenous cultural continuity, carefully considered and managed recreational activities, and learning.</li></ul> | <ul><li>Managers need to be aware of historical processes and future trajectories and be prepared for active management strategies (e.g., assisted migration) to establish and maintain 'future ready' vegetation.</li></ul> | <ul><li>Define and monitor refugia quality, understand potential threats, and investigate how and why species survive in and use these areas within a matrix of differently impacted habitats.</li><li>Immediate identification and adequate protection are critical to the effectiveness of the ecological services provided by temporary refugia.</li></ul> |

Traditional approaches to identifying persistent refugia are based on current-day species diversity and endemism [23]. In Australian rainforests, macroecological approaches have identified refugia by investigating compositional changes in regional communities [24], while more complex models have explored turnover at compositional, functional and phylogenetic levels [25,26]. The objective of preserving evolutionary heritage in persistent refugia is a global challenge that has inspired comparable approaches across diverse habitats [18,20,27–29].

When available, paleo-ecological and fossil evidence can identify and contrast temporal changes of diversity with community persistence at various scales [17,22,30–32]. Such evidence, when combined with landscape genetic and phylogeographic studies from one or more species, can recognize signatures of contraction into, and expansion from refugial areas [15,33,34]. Environmental niche models (ENMs) can then be used to investigate how the distribution of suitable habitat for target species changes through time [35], and in combination with phylogeographic datasets provide validated insights on the geographic location of persistent refugia [36–38].

Environmentally stable areas are likely to preserve high lineage diversity and phylogenetic endemism [26]. However, this may not necessarily result in high levels of speciation into the future, as persistence and speciation represent very different aspects of the evolutionary process [39]. In persistent refugia, the protection of lineage diversity and abundance is an evolutionary outcome often reflecting trait conservatism, whereas high levels of speciation suggest adaptation and/or re-expansion into new or previously vacant habitats [40]. Consequently, persistent refugia are of high biodiversity conservation value and should be protected through legislative action. In order to maintain their key attributes of assembly dynamics within persistent habitats, these refugia need to be safeguarded from disturbances that disrupt their capacity to occur in the multiple stable states that reflect natural, within-community assembly processes [7]. In that case, within-community shifts in species abundances provide a potential measure of the number of ways an ecological community can be rearranged without changing its state including, by definition, the disturbance levels associated with organism senescence, natural regeneration, and community-level competition [6].

### 2.2. Future Refugia

Future refugia may be defined as areas that are predicted to be buffered from anthropogenic climate change and other impacts, thus allowing for environmental and habitat stability and species persistence over time [3]. This definition differs from previous ones that refer more specifically to areas providing spatial and temporal protection from human activities and that will remain suitable for specific taxa in the long-term [14]. The future refugia category revolves around the concept of identifying 'future proofed' vegetation and targeting areas that comprise assemblages of species suited to future climate projections (naturally congregated in situ or established via assisted migration). Protecting future refugia includes managing reserves that are likely to be less impacted by ongoing disturbance and, when the relevant knowledge is available, developing strategies targeting adaptive potential. At the narrowest scale of the concept, planning can be focused on predictive modelling targeting the protection of one or more threatened species [41].

Climate models can distinguish areas less susceptible to future environmental change, or areas that in the predicted future will become suitable for specific vegetation types or selected species [42]. ENM-based approaches incorporating future climate uncertainty across large numbers of species, can also identify areas of high environmental stability inferred from contemporary species distributions fitting the definition of future refugia [43]. For species with little protected habitat within those future-proofed areas, active transplant of putatively adapted genomes might be considered as an option to increase overall viability of local populations [44]. Securing future refugia also involves aligning with land-use planning projections and buffering selected areas from disturbance by protecting and restoring surrounding vegetation [45].

Overall, managing future refugia requires an improved understanding of the adaptive capacity of species obtained through the integration of environmental, physiological, ecological and genetic information [46]. The inclusion of the dynamics of community assembly processes and the potential for multiple stable states [7] in the context of probabilistic rather than deterministic recovery trajectories [6] present some interesting challenges and opportunities for interpretation and experimentation. The challenges inherent in identifying and providing clear definitions for future refugia emerge from the uncertainties in current models for ongoing climatic changes. Identification of future refugia is dependent on the accuracy of predictions around such changes [43].

*2.3. Temporary Refugia*

The temporary refugia category represents current habitat refuges that have not been impacted by recent extreme disturbance events surrounding them (e.g., fire, clearing, grazing, logging, cyclones). Temporary refugia can operate over both shorter and longer time scales and have a critical role in facilitating and favoring localized recovery of surrounding affected areas post disturbance [47]. Temporary refugia might also occur within protected areas at localized scales, and at larger scales in specific landscape contexts (Table 1).

Unlike persistent refugia, the location of temporary refugia can either shift in relation to stochastic extreme events, or be related to specific conditions and therefore be more predictable. A useful example is fire refugia represented by mostly unburned landscape elements within a larger fire-affected landscape matrix [48]. Such unburned areas can support postfire ecosystem functions and biodiversity, as well as provide resilience to further disturbances [49]. They can be small patches spared by fire via localized stochastic events within a continuously burned landscape, or specific areas consistently protected from fire by their location, aspect, structure, and constituent species [50]. Scale, in time and space, can also vary. Within longer-term and spatially broader contexts, temporary refugia can include 'wilderness areas', important areas that have been shown to endure with much reduced species losses and localized extinctions than 'non-wilderness' or managed landscapes [51].

In the current era of increasingly extreme ecosystem disruptions and breakdown, a final and desperate version of temporary refugia includes ex situ refuges such as germplasm collections. These are likely to be less cost effective and more relevant at the species level (with often a focus for threatened or economically important taxa) than at a whole-community level [52].

## 3. The World Heritage Gondwana Rainforests as an Example

The World Heritage Gondwana Rainforests (WHGR) occur in a network of reserves in eastern Australia, from southeast Queensland to just north of Sydney. They are named to reflect their biogeographic origins and ancestry as part of a once more widespread distribution across the super-continent Gondwana [17,53]. The WHGR are part of the 'elevationally restricted mountain ecosystems' placed high on the list of the Australian ecosystems most vulnerable to tipping points [54]. These forests have global conservation and natural heritage significance as defined by the criteria used for assessing outstanding universal value under the World Heritage Convention (UNESCO; World Heritage Criteria vii–x; https://whc.unesco.org/en/list/368/ (accessed on 31 December 2020)). The areas included within the WHGR therefore fit the identification criteria of persistent refugia.

Community assembly and functional diversity studies have identified the distribution and origins of extant diversity [25,55,56]. Multispecies genomic studies have detected the landscape-level signatures of refugial vs. recolonized assemblages [57–59]. Further, ENM-based research has found broad associations between centers of floristic and evolutionary diversity and environmental stability [35]. If used adequately, such knowledge can support the identification of important corridors to enhance connectivity and nat-

ural dispersal potential, areas of diversity that need protection, and the distribution of areas/populations to be targeted for the establishment of future refugia.

Co-occurring species with different trait combinations (e.g., drought tolerance traits reflecting vessel architecture, or dispersal traits such as seed size) can also identify areas of high conservation significance and provide impetus for the development of site- or habitat-specific management strategies [11,60]. This is particularly relevant to the WHGR where functional grouping based on life-history characters can be used to bring together taxa that could respond to selective processes, environmental threats and management actions in similar ways [61]. For instance, it was shown that many threatened rainforest species in the WHGR can 'hold ground' until removed by extreme events, or in some cases become temporally stranded in non-rainforest or marginal environments [62,63]. Developing relatively simple management strategies that consider relevant functional and distributional factors (through targeted translocations and/or localized habitat restoration for example) can go a long way towards ensuring continuing persistence.

The available ecological and evolutionary evidence highlights how in situ management strategies need to include clear guidelines for teams responding to disturbance events and include information about the importance of specific areas and the distribution of key species across vegetation types [64]. Disturbance response guidelines need to be informative and designed to increase awareness about why the species and areas are relevant, and why and how they need protection from specific threats and extreme events. Some species were only found within very limited areas, and so considering alternatives to in situ conservation (e.g., including translocations, assisted migration, and/or germplasm collections) also needs to be part of pre-disturbance planning and post-disturbance responses as immediate actions amid continually changing circumstances.

Using wildfires as an example, human actions related to fuel management and fire suppression are generally prioritized according to a range of non-biological criteria (e.g., human asset protection). The ensuing differences in perception of urgency and type of responses required in the face of extreme fire events can lead to very different outcomes, including for the protection and conservation of native vegetation and habitats. Variations in response to threatening events are difficult to evaluate when not informed by an understanding of the long-term patterns that have shaped landscape vegetation and established refugia [49]. If the objective of biodiversity management is asset-based conservation planning, then explicit goals identifying and quantifying the specific components of biodiversity that are to be protected through dedicated conservation actions need to be defined [64,65]. Target areas for biodiversity conservation need to be identified and protected even if they occur in a matrix of mixed land-use, or in landscapes modified by different disturbance histories [45]. Non-target areas also require some level of protection, especially when they are strategically relevant as buffers [49].

To protect the WHGR refugia from extreme fire events, we need to know and understand the evolutionary history and ecology of what is being protected and why. As an immediate response to the recent fires, burned areas within persistent refugia need to be identified and demarcated. Unburned temporary refugia that can act as source areas for regeneration also need to be identified, protected, and monitored over longer time periods to assess the return of both habitat values and major vegetation structural features. Available knowledge can already be used to protect critical assets by recognizing and implementing buffer zones (e.g., areas of forest protected and allowed to mature) and corridor connections, and by undertaking emergency genetically-informed translocations of target species into safe and climate ready areas that will act as future refugia [62].

## 4. Conclusions

Recent global and regional events demonstrate that refugia should not simply be considered as static conservation elements but placed into a dynamic conservation matrix comprised of management options that consider both the static persistence and stochastic emergence of refugial areas [3].

In the face of anthropogenic climate change and recurring extreme events, we are faced with hard decisions. Should we protect persistent refugia as a priority, or should we consider ex situ strategies such as germplasm collections or translocation into future refugia for threatened and other species? Despite the immediate attraction of some of these ideas, there needs to be scrutiny and a long-term vision for those strategies to succeed. What is the likelihood of success? How much of the available evolutionary potential should and can be stored/translocated [66]? Where, when, and how is the material preserved in germplasm refuges going to be used in the long-term?

Knowing how persistent refugia have enhanced the survival of diversity over protracted temporal scales will inform the management decisions that protect species into an uncertain future. We urgently need more information about the ecological conditions that potentially precipitate a state-change in any type of refugia, and about the evolutionary, population dynamic, and ecological responses induced by such changes at different spatial and temporal scales. Only with that information will it be possible to plan long- and short-term management responses to such threats and develop meaningful monitoring strategies that can evaluate success.

**Author Contributions:** M.R. and R.K. were equally involved in developing the concepts and writing the manuscript. All authors have read and agreed to the published version of the manuscript.

**Funding:** This research was funded by the Royal Botanic Gardens and Domain Trust.

**Acknowledgments:** We would like to acknowledge the valuable comments and extremely helpful suggestions provided by the editor and two anonymous reviewers, the NSW Department of Planning, Industry and Environment for identifying the question, and Brett Summerell for providing comments on an early version of the manuscript.

**Conflicts of Interest:** The authors declare no conflict of interest.

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
