# Peer review of "Conserving Refugia: What Are We Protecting and Why?"

_diversity, doi:10.3390/d13020067_

Round 1

Reviewer 1 Report

The manuscript under review titled as “Conserving refugia – what are we protecting and why? “ by Maurizio Rossetto and Robert Kooyman opens an important discussion on new concepts and possible improvements in protection of natural diversity. The value of refugia for conservation of particularly rich and phylogenetically diverse areas has long been recognized. Authors of this manuscript, however, suggest a more systematic approach to protection of biodiversity preserved in refugia. Importantly, they suggest differentiation of refugia into three categories: Persistent, Transient, and Future refugia. This differentiation can potentially allow a more efficient treatment of urgent issues of practical conservation. The suggested approach stresses the importance of preservation of evolutionary heritage. It also lays foundation for a prospective treatment of communities in protected areas, accounting for anticipated changes in current environments of these areas.

This conceptual study is timely and thoroughly justified. In my view, Plant Diversity will benefit from this publication on its pages. I therefore suggest to accept this manuscript for publication. There are several minor issues, however, which should be resolved before the final acceptance.

In the Abstract, the following sentence needs specification, as suggested (in italics): “As large natural landscapes worldwide are declining and increasingly threatened by extreme events, critical decision-making in biological conservation depends on improved understanding of what is being protected by refugia and why.”

In the Introduction, particularly the chapter “Management-oriented definitions of refugia” should account for idiosyncrasy and specificity of responses by different species to environmental changes. The same change may promote very different responses in different species. This problem has not been properly presented (or discussed later).

As regards Table 1, I would like to suggest a more active and concise style of presentation of Main Identification Criteria. For example, for the Persistent Refugia, I suggest the following style:

Identify:

distinctiveness at compositional, functional and phylogenetic levels

community assembly processes and persistence – by palaeoecological and fossil evidence

signals of contraction / expansion dynamics – by landscape genomic and phylogeographic studies

areas of high environmental stability – by ENMs

A similar style should be adapted for definitions of the other two categories.

Definition of Temporary Refugia in the table is a bit vague. For example, the following definition criterium - Refuges can involve longer-term, spatially larger areas such as ‘wilderness areas’, or whole National Parks - does not allow to differentiate between Temporary and Persistent Refugia. A clearer separation between these two categories would be preferable.

In description of Persistent Refugia, the conceptual advances by the group of Prof. Andreas Prinzing from France should be mentioned. For example, their results on the role of habitat types and plant communities they harbor in maintaining adaptations to the biotic interactions and abiotic environments of past epochs (Gerhold et al. 2015, Prinzing et al. 2017; Bartish et al. 2020). The ideas developed by these studies suggest that refugia not only protect diversity of taxa and lineages, they also protect specific interactions and adaptations to stable environments of the ancient epochs.

Discussion on endemism (lines 96-97) should introduce briefly the concept of phylogenetic endemism, developed by Rosauer et al. (2009).

The chapter on Future Refugia should account for uncertainties in current models for further climatic changes. Identification of Future Refugia depends on accuracy of prediction of possible changes. This is an area of a very hot political debate. This debate should be acknowledged.

In definition and discussion of the Temporary Refugia, the difference from Persistent Refugia should be clearly stated.

I suggest adding a paragraph on discussion of similar approaches, which focused on preservation of evolutionary heritage by habitat types of Persistent Refugia on different continents. For example, in Africa (Forest et al. 2007) and Europe (Winter et al. 2012; Bartish et al. 2020), and possibly on other continents. This paragraph could add a stronger global perspective to your study.

Finally, in my personal view, the proportion of self-citations exceeding 25% is a bit too large. Are all of these references really necessary? If you are certain they are, this would be fine to me.

References

Bartish, I. V., et al. (2020) Anthropogenic threats to evolutionary heritage of angiosperms in the Netherlands through an increase in high-competition environments. Conservation Biology 34(6): 1536-1548.

Forest, F., et al. (2007) Preserving the evolutionary potential of floras in biodiversity hotspots. Nature 445(7129): 757-760.

Gerhold, P., et al. (2015) Phylogenetic patterns are not proxies of community assembly mechanisms (they are far better). Functional Ecology 29(5): 600-614.

Prinzing, A., et al. (2017) Benefits from living together? Clades whose species use similar habitats may persist as a result of eco-evolutionary feedbacks. New Phytologist 213(1): 66-82.

Rosauer, D. et al. (2009) Phylogenetic endemism: a new approach for identifying geographical concentrations of evolutionary history. Molecular Ecology 18(19): 4061-4072.

Winter, M., et al. (2013) Phylogenetic diversity and nature conservation: where are we? Trends in Ecology & Evolution 28(4): 199-204.

Author Response

The manuscript under review titled as “Conserving refugia – what are we protecting and why?“ by Maurizio Rossetto and Robert Kooyman opens an important discussion on new concepts and possible improvements in protection of natural diversity. The value of refugia for conservation of particularly rich and phylogenetically diverse areas has long been recognized. Authors of this manuscript, however, suggest a more systematic approach to protection of biodiversity preserved in refugia. Importantly, they suggest differentiation of refugia into three categories: Persistent, Transient, and Future refugia. This differentiation can potentially allow a more efficient treatment of urgent issues of practical conservation. The suggested approach stresses the importance of preservation of evolutionary heritage. It also lays foundation for a prospective treatment of communities in protected areas, accounting for anticipated changes in current environments of these areas.

This conceptual study is timely and thoroughly justified. In my view, Plant Diversity will benefit from this publication on its pages. I therefore suggest to accept this manuscript for publication. There are several minor issues, however, which should be resolved before the final acceptance.

Response: Thank you for the very helpful feedback that considerably improved the paper.

In the Abstract, the following sentence needs specification, as suggested (in italics): “As large natural landscapes worldwide are declining and increasingly threatened by extreme events, critical decision-making in biological conservation depends on improved understanding of what is being protected by refugia and why.”

Response: Added as suggested.

In the Introduction, particularly the chapter “Management-oriented definitions of refugia” should account for idiosyncrasy and specificity of responses by different species to environmental changes. The same change may promote very different responses in different species. This problem has not been properly presented (or discussed later).

Response: updated text as suggested.

As regards Table 1, I would like to suggest a more active and concise style of presentation of Main Identification Criteria. For example, for the Persistent Refugia, I suggest the following style:

Identify: distinctiveness at compositional, functional and phylogenetic levels ; community assembly processes and persistence – by palaeoecological and fossil evidence ; signals of contraction / expansion dynamics – by landscape genomic and phylogeographic studies ; areas of high environmental stability – by ENMs

Response: Table updated as suggested.

A similar style should be adapted for definitions of the other two categories.

Definition of Temporary Refugia in the table is a bit vague. For example, the following definition criterium - Refuges can involve longer-term, spatially larger areas such as ‘wilderness areas’, or whole National Parks - does not allow to differentiate between Temporary and Persistent Refugia. A clearer separation between these two categories would be preferable.

In description of Persistent Refugia, the conceptual advances by the group of Prof. Andreas Prinzing from France should be mentioned. For example, their results on the role of habitat types and plant communities they harbor in maintaining adaptations to the biotic interactions and abiotic environments of past epochs (Gerhold et al. 2015, Prinzing et al. 2017; Bartish et al. 2020). The ideas developed by these studies suggest that refugia not only protect diversity of taxa and lineages, they also protect specific interactions and adaptations to stable environments of the ancient epochs.

Response: updated text as suggested.

Discussion on endemism (lines 96-97) should introduce briefly the concept of phylogenetic endemism, developed by Rosauer et al. (2009).

Response: updated text as suggested.

The chapter on Future Refugia should account for uncertainties in current models for further climatic changes. Identification of Future Refugia depends on accuracy of prediction of possible changes. This is an area of a very hot political debate. This debate should be acknowledged.

Response: updated text as suggested.

In definition and discussion of the Temporary Refugia, the difference from Persistent Refugia should be clearly stated.

I suggest adding a paragraph on discussion of similar approaches, which focused on preservation of evolutionary heritage by habitat types of Persistent Refugia on different continents. For example, in Africa (Forest et al. 2007) and Europe (Winter et al. 2012; Bartish et al. 2020), and possibly on other continents. This paragraph could add a stronger global perspective to your study.

Response: updated text as suggested.

Finally, in my personal view, the proportion of self-citations exceeding 25% is a bit too large. Are all of these references really necessary? If you are certain they are, this would be fine to me.

Response: Good point, however, much of the local, regional, continental perspective is backed by our work. The addition of broader perspectives (global) and the increase in references sees this issue somewhat reduced.

Reviewer 2 Report

Dear authors,

Your manuscript is really interesting - I have provided some comments and suggestions that I hope you will find useful and constructive

Author Response

Your manuscript is really interesting - I have provided some comments and suggestions that I hope you will find useful and constructive.

Response: Thank you for the very helpful feedback that considerably improved the paper.

L32: Please provide a reference for this statement.

Response: Done.

L44 and L52: Please consider mentioning the following reference, as it is directly linked to your study and state how it's different from what you are doing/proposing. (Keppel, G., et al. 2018).

Response: Done with relevant text added as well.

L60: Do you mean diversification?

Response: Clarified.

L63: Characterized?

Response: Changed.

L65: How are future refugia different from the definition given by Monsarrat et al. (2019)? You should mention this reference as it is directly linked to your study.

Response: updated as suggested.

Table 1: Please provide an in-text explanation before Table 1 about the main and core value criteria, as well as the management strategy criteria. What's the difference between the main and the core value criteria definition?

Response: clarified within text. All other comments within the table were attended to as well.

L73: You should also mention Laffan et al. (2010) who provide a method for detecting paleo- and neo-endemism hotspots, that are directly linked to 'persistent' refugia.

Response: updated as suggested. Additional comments in first para also attended to in table and response to previous reviewer.

L95: You should mention here several studies dealing with the location of paleo-endemic refugia via lineage-specific ENMs. Rosauer et al. (2015); Carnaval et al. (2014).  

Response: Done.

L97: You should mention the following reference and discuss how the problems they raise can be overcome. Cantalapiedra et al. (2019).

Response: Additional text and refs added.

L107: As mentioned earlier, you should make clear how your definition is different from that introduced by Monsarrat et al. 2019.

Response: clarified within text of first para.

L134: refugial.

Response: to provide better clarity, refuge was changed to refugia across the paragraph.

Round 2

Reviewer 2 Report

Dear authors,

Thank you for addressing all the issues raised and for delivering a very interesting manuscript that I enjoyed reading.